# Meta-analysis of effects of exclusive breastfeeding on infant gut microbiota across populations

Nhan T. Ho[1], Fan Li[2], Kathleen A. Lee-Sarwar[3,4], Hein M. Tun[5,6], Bryan P. Brown[7,8,15], Pia S. Pannaraj[9], Jeffrey M. Bender[9], Meghan B. Azad[10], Amanda L. Thompson[11], Scott T. Weiss[4], M. Andrea Azcarate-Peril[12,13], Augusto A. Litonjua[14], Anita L. Kozyrskyj[5], Heather B. Jaspan[8,15], Grace M. Aldrovandi[2] & Louise Kuhn[1]

Previous studies on the differences in gut microbiota between exclusively breastfed (EBF) and non-EBF infants have provided highly variable results. Here we perform a meta-analysis of seven microbiome studies (1825 stool samples from 684 infants) to compare the gut microbiota of non-EBF and EBF infants across populations. In the first 6 months of life, gut bacterial diversity, microbiota age, relative abundances of Bacteroidetes and Firmicutes, and predicted microbial pathways related to carbohydrate metabolism are consistently higher in non-EBF than in EBF infants, whereas relative abundances of pathways related to lipid metabolism, vitamin metabolism, and detoxification are lower. Variation in predicted microbial pathways associated with non-EBF infants is larger among infants born by Caesarian section than among those vaginally delivered. Longer duration of exclusive breastfeeding is associated with reduced diarrhea-related gut microbiota dysbiosis. Furthermore, differences in gut microbiota between EBF and non-EBF infants persist after 6 months of age. Our findings elucidate some mechanisms of short and long-term benefits of exclusive breastfeeding across different populations.

[1] Gertrude H. Sergievsky Center, Columbia University, New York City, NY 10032, USA. [2] Department of Pediatrics, University of California, Los Angeles, CA 90095, USA. [3] Division of Rheumatology, Immunology and Allergy, Department of Medicine, Brigham and Women's Hospital and Harvard Medical School, Boston, MA 02115, USA. [4] Channing Division of Network Medicine, Department of Medicine, Brigham and Women's Hospital and Harvard Medical School, Boston, MA 02115, USA. [5] Department of Pediatrics, Faculty of Medicine and Dentistry, University of Alberta, Edmonton, T6G 1C9 AB, Canada. [6] HKU-Pasteur Research Pole, School of Public Health, Li Ka Shing Faculty of Medicine, The University of Hong Kong, Hong Kong SAR, China. [7] Duke University, Durham, NC 27708, USA. [8] University of Cape Town Health Sciences Faculty, Institute of Infectious Disease and Molecular Medicine, Cape Town, 7701, South Africa. [9] Children's Hospital Los Angeles, University of Southern California, Los Angeles, CA 90027, USA. [10] Children's Hospital Research Institute of Manitoba, Department of Pediatrics & Child Health, University of Manitoba, Winnipeg, R3E 3P4 Manitoba, Canada. [11] Department of Anthropology, University of North Carolina, Chapel Hill, NC 27599, USA. [12] Department of Medicine, Division of Gastroenterology and Hepatology, School of Medicine, University of North Carolina, Chapel Hill, NC 27599, USA. [13] Microbiome Core Facility, Center for Gastrointestinal Biology and Disease, School of Medicine, University of North Carolina, Chapel Hill, NC 27599, USA. [14] Division of Pulmonary Medicine, Department of Pediatrics, University of Rochester Medical Center, Rochester, NY 14642, USA. [15] Seattle Children's Research Institute, University of Washington, Seattle, WA 98101, USA. Correspondence and requests for materials should be addressed to L.K. (email: lk24@columbia.edu)

Establishment of the gut microbiota in early life has substantial impact on subsequent health[1]. Common sources of the infant's intestinal microorganisms are from the mother's skin, vagina, stool, and from breastfeeding[2–5]. There is a close relationship between the infant's gut microbiota and the mother's breast milk microbiota and human milk oligosaccharides (HMO) composition[6–9]. Indeed, recent evidence has shown that breast milk microbiota can directly seed the infant gut microbiota, and the effects of breastmilk on infant gut microbiota are dose-dependent[5]. The microbiota in breast milk changes over time during lactation and has been shown to be different between exclusive breastfeeding (EBF) and non-EBF mothers[10,11]. Gut microbial abundances in breastfed infants, especially bifidobacterial species, are correlated with the mother's HMOs and HMO-related catabolic activity[3,12,13]. Infant gut microbiota have been shown to be different between breastfed and formula-fed infants[14–20] and change rapidly during the transition from breastfeeding to formula[21].

EBF in the first 6 months of life provides a multitude of health benefits[22]. For example, EBF has been shown to be strongly protective against diarrhea, morbidity, and mortality[23] and decreases long-term risk of diabetes and obesity as compared to non-EBF or formula-fed infants[24–26]. We hypothesize that the numerous benefits of EBF may be in part due to its effects on the infant gut microbiota. Several recent studies have identified varying differences in gut microbial composition or diversity between EBF and non-EBF infants[5,27–29] or gradients in the gut microbiota composition or diversity across EBF, non-EBF, and non-breastfed (non-BF) infants[5,14,28,30,31]. However, some other studies have found no significant differences in gut microbial communities between EBF and non-EBF infants[3,32]. In addition, mode of delivery has been variably reported to have no effect[33] or a significant effect[29] or a potential interaction effect with breastfeeding[30] on the infant gut microbiota. The wide variation in reported results together with heterogeneity in feeding category definitions, study designs, study populations, and especially in data processing and analysis methods make these findings difficult to synthesize and interpret.

In this study, we apply robust statistical methods to analyze gut microbiome data and perform meta-analyses pooling estimates from seven published microbiome studies (a total of 1825 stool samples from 684 infants) to investigate the effects of EBF compared with non-EBF on infant gut microbiota across different populations. We find that in the first 6 months of life, overall bacterial diversity, gut microbiota age, relative abundances of Bacteroidetes and Firmicutes, and microbial-predicted pathways related to carbohydrate metabolism are consistently increased in non-EBF vs. EBF infants. In contrast, relative abundances of pathways related to lipid, vitamin metabolism, and detoxification are decreased in non-EBF vs. EBF infants. The perturbation in microbial-predicted pathways associated with non-EBF is larger in infants delivered by C-section than in those delivered vaginally. Longer duration of EBF is associated with reduced diarrhea-related gut microbiota dysbiosis, and the effects of EBF persist after 6 months of age. Taken together, these consistent findings across vastly different populations suggest that alteration in gut microbiota and their functional pathways may represent a key mechanism for the short-term and long-term benefits of EBF.

## Results

### Microbial diversity is increased in non-EBF vs. EBF infants.
In infants ≤6 months of age, across the seven included studies, non-EBF is consistently associated with increased gut microbial alpha diversity (standardized Shannon index) compared with EBF adjusting for infant age at sample collection (pooled standardized diversity difference [DD] = 0.34 standard deviation [sd], 95% confidence interval [95% CI] = [0.20; 0.48], random effects model pooled $p$-value < 0.001; Fig. 1a, b). In a subset of five studies that also contained non-BF infants, including Bangladesh[34], Canada[30], USA (California–Florida [CA–FL][5], USA (California–Massachusetts–Missouri [CA–MA–MO])[18], and USA (North Carolina [NC])[28], gut microbiome diversity (standardized Shannon index) was significantly increased in infants with less breastfeeding after adjusting for age of infants at sample collection (pooled standardized DD = 0.39 sd, 95% CI = [0.19; 0.58], random effects model pooled $p$-value < 0.001; Fig. 1c). Results were consistent utilizing three other commonly used alpha diversity indices (phylogenetic diversity whole tree, observed species, Chao1; all random effects model pooled $p$-values < 0.05; Fig. 1d, e).

In sensitivity meta-analyses excluding estimates from either the USA (NC) study[28] (which contained a small number of infants ≤6 months old) or the Haiti study[3] (which included samples from HIV-uninfected infants born to HIV-infected and HIV-uninfected mothers) or the Vitamin D Antenatal Asthma Reduction Trial (VDAART) study in the USA (CA–MA–MO)[18] (which contained samples from infants at high risk of asthma and allergies, half of whom were randomized to high-dose antenatal vitamin D supplementation), the results remained similar (Supplementary Fig. 1).

In a subset of four studies (Canada[30], Haiti[3], USA [CA–FL][5], and USA [CA–MA–MO][18]) with available data on mode of delivery, the increase in microbial diversity associated with non-EBF was similar in the meta-analysis stratified on vaginally delivered infants and cesarean-delivered infants (Supplementary Fig. 2).

In a subset of four studies with available data on infant sex (Bangladesh[34], Haiti[3], USA [CA–FL][5], USA [NC][28]), the increase in microbial diversity associated with non-EBF was similar in the analysis adjusting for infant age and in the analysis adjusting for both infant age and sex (Supplementary Fig. 3).

### Microbiota age is increased in non-EBF vs. EBF infants.
A Random Forest (RF) model was used to predict the infant age in each included study based on relative abundances of the shared gut bacterial genera of the seven included studies (Supplementary Table 1). The model explained 95% of the variance related to chronologic age in the training set and 65% of the variance related to chronologic age in the test set of Bangladesh data (Supplementary Fig. 4). The predicted infant age in each included study based on relative abundances of the shared gut bacterial genera using this RF model was regarded as gut microbiota age.

In infants ≤6 months of age, a consistent (6/7 studies) increase in gut microbiota age was observed in non-EBF as compared to EBF infants after adjusting for infant age at sample collection (pooled standardized microbiota age difference [MD] = 0.33 sd, 95% CI = [0.09; 0.58], random-effects model pooled $p$-value = 0.007; Fig. 2a, b). In sensitivity meta-analyses excluding either of the three studies mentioned above[3,18,28], the association remained similar (Supplementary Fig. 5). In the subset of four studies with available data on mode of delivery[3,5,18,30], meta-analysis stratified on vaginally delivered infants and on cesarean-delivered infants showed similar overall increase in microbiota age in non-EBF vs. EBF infants (Supplementary Fig. 6).

The trend of increasing gut microbiota age in infants with less breastfeeding after adjusting for age of infants at sample collection was also observed in a subset of five studies containing a non-BF group[5,18,28,30,34] (pooled standardized MD = 0.35 sd, 95% CI = [0.09; 0.61], random-effects model pooled $p$-value = 0.008; Fig. 2c).

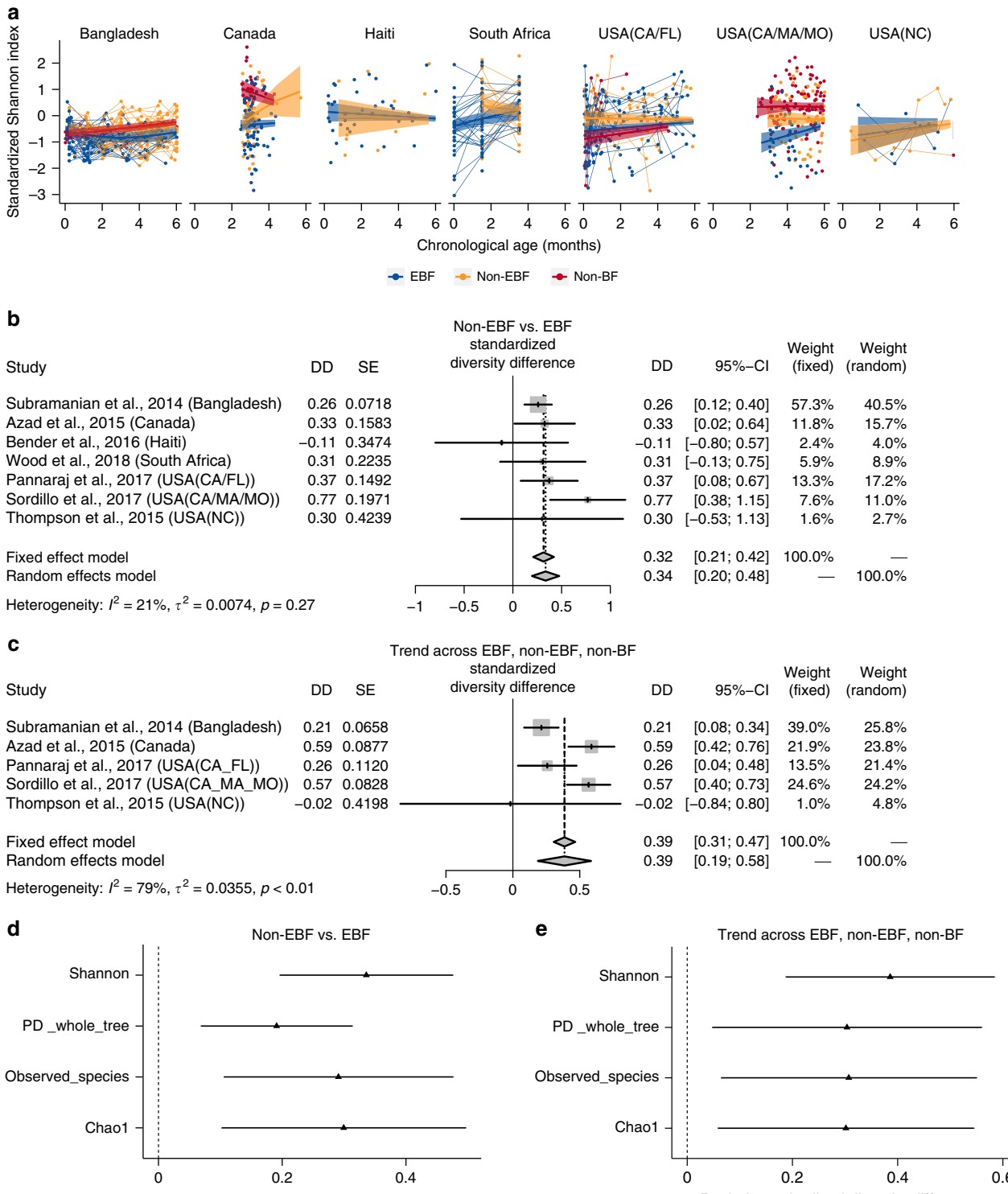

In a subset of four studies with available data on infant sex[3,5,28,34], the change in microbiota age associated with non-EBF was similar in the analysis adjusting for infant age and in the analysis adjusting for both infant age and sex (Supplementary Fig. 7).

**Microbial composition is altered in non-EBF vs. EBF infants.** Across the seven included studies, there was a large heterogeneity in the difference in log odds of gut bacterial taxa relative abundances between non-EBF and EBF infants after adjusting for age

of infants at sample collection. Notably, a decrease in relative abundance of Proteobacteria in non-EBF vs. EBF infants was observed in the four studies in North America, but the opposite was observed in the other three studies in Haiti, South Africa, and Bangladesh (Fig. 3a). However, there was also remarkable consistency across studies. At the phylum level, there was an overall significant increase in the relative abundances of Bacteroidetes (consistent in all seven studies) and Firmicutes (consistent in 6/7 studies) in non-EBF vs. EBF infants (all random-effects model pooled $p$-values < 0.05 and false discovery rate (FDR)-adjusted

**Fig. 1** Effects of non-EBF vs. EBF on gut microbial diversity in infants ≤6 months of age. **a** Gut microbial alpha diversity (standardized Shannon index) by breastfeeding status by infant age at stool sample collection from each included studies. Fitted lines and 95% confidence intervals (95% CI) were from generalized additive mixed models (GAMM). **b** The difference in gut microbial alpha diversity (standardized Shannon index) between non-EBF and EBF infants ≤6 months of age from each study and the pooled effect across seven included studies (meta-analysis) with 95% CI. **c** The trend effect of gut microbial alpha diversity (standardized Shannon index) across EBF, non-EBF, and non-BF infants ≤6 months of age from each study and the pooled effect across five included studies (meta-analysis) with 95% CI. Data from Haiti and South Africa studies were not included as there was no non-BF group. In each study, to roughly test for trends across breastfeeding categories, breastfeeding was coded as a continuous variable (EBF = 1, non-EBF = 2, and non-BF = 3). **d** Pooled estimates and 95% CI for the difference in (standardized) gut microbial alpha diversity (four common alpha diversity indexes) between non-EBF and EBF infants ≤6 months of age. **e** Pooled estimates and 95% CI for the trend effect of (standardized) gut microbial alpha diversity (four common alpha diversity indexes) across EBF, non-EBF, and non-BF infants ≤6 months of age. Estimates for diversity difference or trend and corresponding standard errors from each study were from linear mixed-effect models (longitudinal data) or linear models (non-longitudinal data) and were adjusted for infant age at sample collection. Pooled estimates of standardized diversity difference or trend and their 95% CI were from random-effects meta-analysis models based on the adjusted estimates and corresponding standard errors of all included studies. Pooled estimates with false discovery rate (FDR)-adjusted pooled p-values < 0.1 are shown as triangles. EBF exclusive breastfeeding, non-EBF non-exclusive breastfeeding, non-BF non-breastfeeding, USA United States of America, CA California, FL Florida, MA Massachusetts, MO Missouri, NC North Carolina, DD diversity difference, SE standard error, PD_whole_tree phylogenetic diversity whole tree

---

pooled p-values < 0.1; Fig. 3a, Supplementary Table 2). There was also a consistent trend of increasing relative abundances of these two phyla across EBF, non-EBF, and non-BF in the subset of five studies with a non-BF group[5,18,28,30,34] (all random-effects model pooled p-values < 0.05 and FDR-adjusted pooled p-values < 0.1; Supplementary Table 3).

At the order level, the relative abundances of Bacteriodales (7/7 studies) and Clostridiales (6/7 studies) were consistently increased in non-EBF infants (all random-effects model pooled p-values < 0.05 and FDR-adjusted pooled p-values < 0.1; Supplementary Fig. 8, Supplementary Table 2). At the family level, the relative abundances of Bacteroidaceae (7/7 studies) and Veillonellaceae (6/7 studies) were consistently increased in non-EBF infants (all random-effects model pooled p-values < 0.05 and FDR-adjusted pooled p-values < 0.1; Fig. 3b, Supplementary Table 2). At the genus level, there were increases in the relative abundances of Bacteroides (7/7 studies), Eubacterium, Veillonella (6/7 studies), and Megasphaera (5/7 studies) in non-EBF infants (all random-effects model pooled p-values < 0.05, FDR-adjusted pooled p-value of Eubacterium < 0.1; Supplementary Fig. 9, Supplementary Table 2).

In sensitivity meta-analyses excluding either of the three studies mentioned above[3,18,28], the results remained similar (Supplementary Tables 4, 5, 6).

In the subset of four studies with available data on mode of delivery[3,5,18,30], the results of meta-analysis stratified on vaginally delivered infants were similar to those of meta-analysis stratified on cesarean-delivered infants from the phylum to family level. Phylum Proteobacteria (particularly family Enterobacteriaceae) was markedly and significantly reduced in non-EBF infants, especially among infants delivered by cesarean (Supplementary Figs. 10, 11, 12, Supplementary Table 7). At the genus level, relative abundance of Acidaminococcus was significantly higher in vaginally delivered non-EBF vs. vaginally delivered EBF infants, whereas relative abundances of Proteus and Anaerotruncus were significantly lower in cesarean-delivered non-EBF vs. cesarean-delivered EBF infants (all FDR-adjusted pooled p-values < 0.1; Supplementary Figs. 13, 14, Supplementary Table 7).

In a subset of four studies with available data on infant sex[3,5,28,34], the change in gut microbial composition at the phylum level associated with non-EBF was similar in the analysis adjusting for infant age and in the analysis adjusting for both infant age and sex (Supplementary Fig. 15).

**Microbial functions are altered in non-EBF vs. EBF infants**. Across the seven included studies, although the difference in log odds of relative abundances of gut bacterial KEGG functional pathways between non-EBF and EBF infants was largely heterogeneous, important consistencies were found. At KEGG level 2, there was no pathway significantly different between non-EBF and EBF group after adjusting for multiple testing (Supplementary Fig. 16 and Supplementary Table 8). At KEGG level 3, the relative abundances of 24 pathways were significantly different between non-EBF and EBF infants (random-effects model pooled p-values < 0.05), and eight of these remained significant after adjusting for multiple testing (FDR-adjusted pooled p-values < 0.1; Fig. 4a). Specifically, after adjusting for age of infants at sample collection, in the non-EBF group vs. the EBF group, there were higher relative abundances of several pathways related to carbohydrate metabolism, viz. fructose and mannose metabolism, pentose and glucuronate interconversions, and pentose phosphate pathway, as well as fatty acid biosynthesis and biosynthesis of ansamycins pathways. In addition, in non-EBF infants, there were lower relative abundances of some pathways related to lipid metabolism, lipid homeostasis, and free radical detoxification (fatty acid metabolism and peroxisome), and vitamin 6 metabolism (all random-effects model pooled p-values < 0.05 and FDR-adjusted pooled p-values < 0.1; Fig. 4a, Supplementary Table 8). Sensitivity meta-analyses excluding either of the three studies mentioned above[3,18,28] showed similar results (Supplementary Table 9).

In the subset of four studies with available data on mode of delivery[3,5,18,30], meta-analysis stratified by mode of delivery showed remarkable heterogeneity between vaginally delivered infants and cesarean-delivered infants. In vaginally delivered infants, there were six pathways related to carbohydrate and lipid metabolism significantly perturbed between non-EBF and EBF infants after adjusting for multiple testing (all FDR-adjusted pooled p-values < 0.1; Fig. 4b, Supplementary Table 10). Whereas, in cesarean-delivered infants only, there were a much larger number of pathways (35 pathways) of many cellular and metabolic processes (such as cell growth and death, membrane transport, replication and repair, carbohydrate, lipid, amino acid, vitamin, and energy metabolism) significantly perturbed between non-EBF and EBF infants after adjusting for multiple testing (all FDR-adjusted pooled p-values < 0.1). In addition, the perturbation of these pathways was mostly consistent across four included studies (Fig. 4c, Supplementary Table 10).

In a subset of four studies with available data on infant sex[3,5,28,34], there were four KEGG pathways at level 3 significantly perturbed between non-EBF and EBF infants after adjusting for multiple testing in the analysis adjusting for infant age, and one pathway remained significant in the analysis adjusting for both infant age and sex (Supplementary Fig. 17).

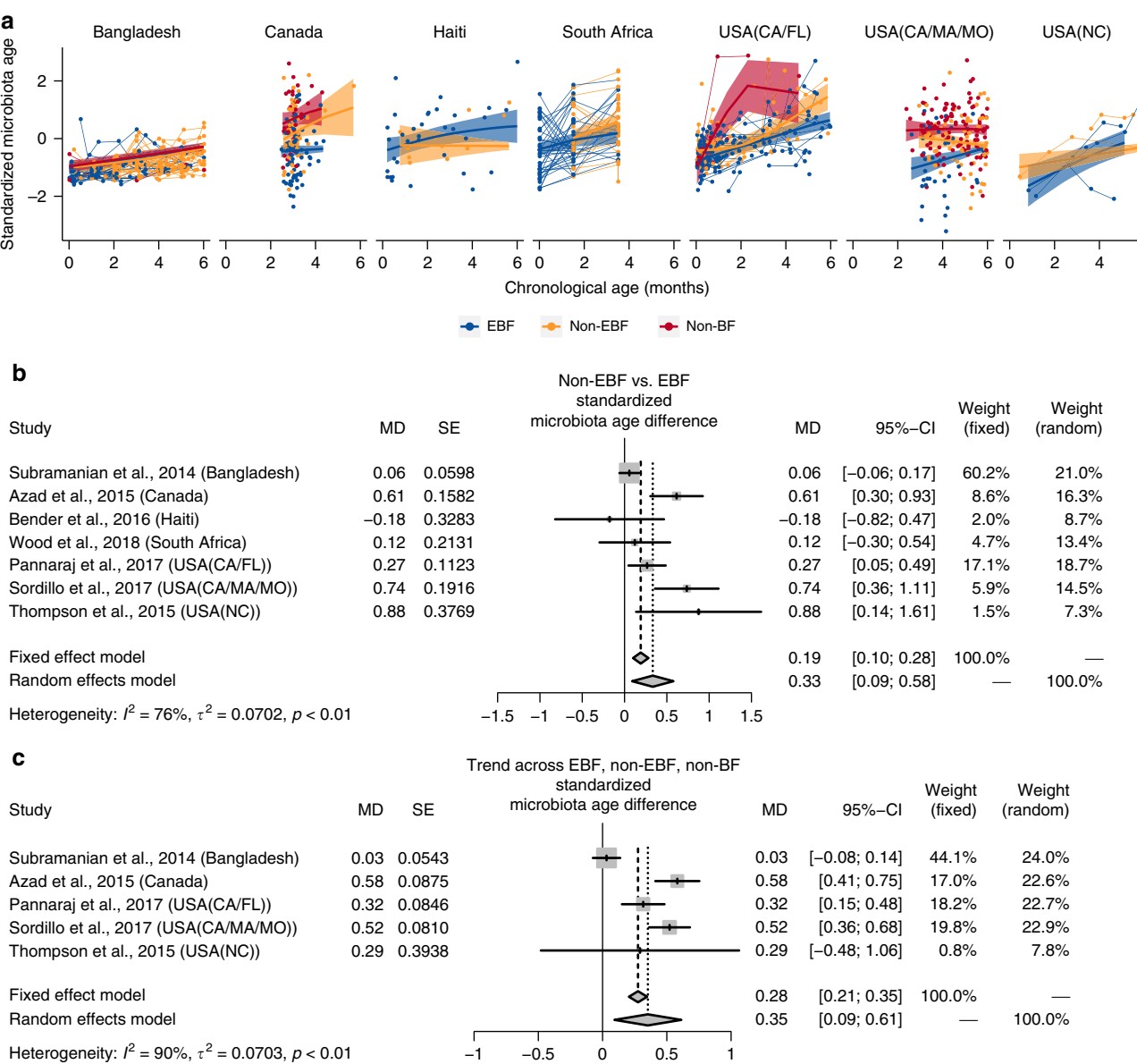

**Fig. 2** Effects of non-EBF vs. EBF on gut microbiota age in infants ≤6 months of age. **a** Gut (standardized) microbiota age of infants ≤6 months of age by breastfeeding status by age of infants at stool sample collection from each of seven included studies. Fitted lines and 95% confidence intervals (95% CI) were from generalized additive mixed models (GAMM). **b** The difference in gut (standardized) microbiota age between non-exclusively breastfed (non-EBF) and EBF infants ≤6 months of age from each study and the pooled effect across seven included studies (meta-analysis) with 95% CI. **c** The trend of gut (standardized) microbiota age across EBF, non-EBF, and non-BF infants ≤6 months of age from each study and the pooled effect across five included studies (meta-analysis) with 95% CI. The Haiti and South Africa studies were not included, as there was no non-BF group in these two studies. In each study, to test for trend across breastfeeding categories, breastfeeding was coded as a continuous variable in the model (EBF = 1, non-EBF = 2, and non-BF = 3). Estimates for (standardized) microbiota age difference or trend and corresponding standard error from each study were from linear mixed-effect models (for longitudinal data) or linear models (for non-longitudinal data) and were adjusted for age of infants at sample collection. EBF exclusive breastfeeding, non-EBF non-exclusive breastfeeding, non-BF no breastfeeding, USA United States of America, CA California, FL Florida, MA Massachusetts, MO Missouri, NC North Carolina, MD microbiota age difference, SE standard error

**Duration of EBF and gut microbiota differences after 6 months of age**. Using data from the Bangladesh study only, which included 996 stool samples collected monthly from birth to 2 years of life in 50 subjects, we found that shorter duration of EBF (less than 2 months vs. more than 2 months from birth) was associated with a larger increase in gut microbiota age. This association was significant from 6 months to 15 months of age (MD = 1.64 months, 95% CI = [0.23, 3.05], p-value = 0.02; Fig. 5a).

From 6 months to 2 years of age, after adjusting for age of infants at sample collection, shorter duration of EBF (less than 2 months vs. more than 2 months from birth) was associated with lower relative abundance of the phylum Actinobacteria and higher relative abundance of Firmicutes (all FDR-adjusted p-values < 0.1; Supplementary Table 11). At the family level, infants with shorter duration of EBF had lower relative abundances of Bifidobacteriaceae and Enterococcaceae and higher relative abundances of Lactobacillaceae, Coriobacteriaceae, Prevotellaceae, Clostridiaceae, Erysipelotrichaceae, and Lachnospiraceae (all FDR-adjusted p-values < 0.1; Fig. 5b, Supplementary Table 11).

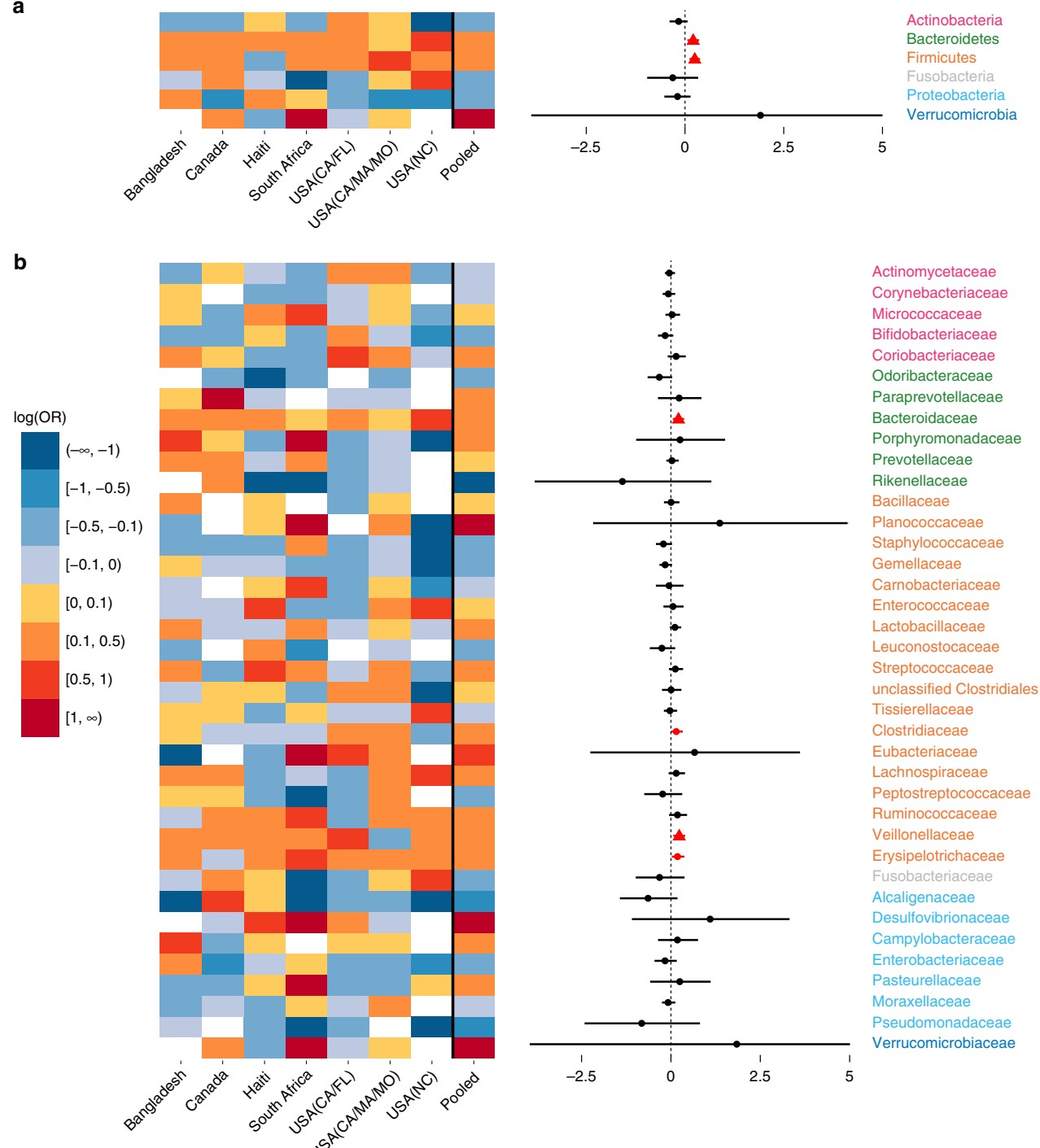

**Fig. 3** Effects of non-EBF vs. EBF on gut bacterial taxa abundances in infants ≤6 months of age. **a** Gut bacterial phyla: heatmap of log(odds ratio) (log[OR]) of relative abundances of all gut bacterial phyla between non-EBF and EBF infants for each study and forest plot of pooled estimates across all seven studies with 95% confidence intervals (95% CI). **b** Gut bacterial families: heatmap of log(OR) of relative abundances of all gut bacterial families between non-EBF and EBF infants for each study and forest plot of pooled estimates across all seven studies with 95% CI. All log(OR) estimates of each bacterial taxa from each study were from generalized additive models for location scale and shape (GAMLSS) with zero-inflated beta family (BEZI) and were adjusted for age of infants at sample collection. Pooled log(OR) estimates and 95% CI (forest plot) were from random-effects meta-analysis models based on the adjusted log(OR) estimates and corresponding standard errors of all included studies. Pooled log(OR) estimates with pooled p-values < 0.05 are in red, and those with false discovery rate (FDR)-adjusted pooled p-values < 0.1 are shown as triangles. Missing (unavailable) values are in white. EBF exclusive breastfeeding, non-EBF non-exclusive breastfeeding, OR odds ratio, USA United States of America, CA California, FL Florida, MA Massachusetts, MO Missouri, NC North Carolina

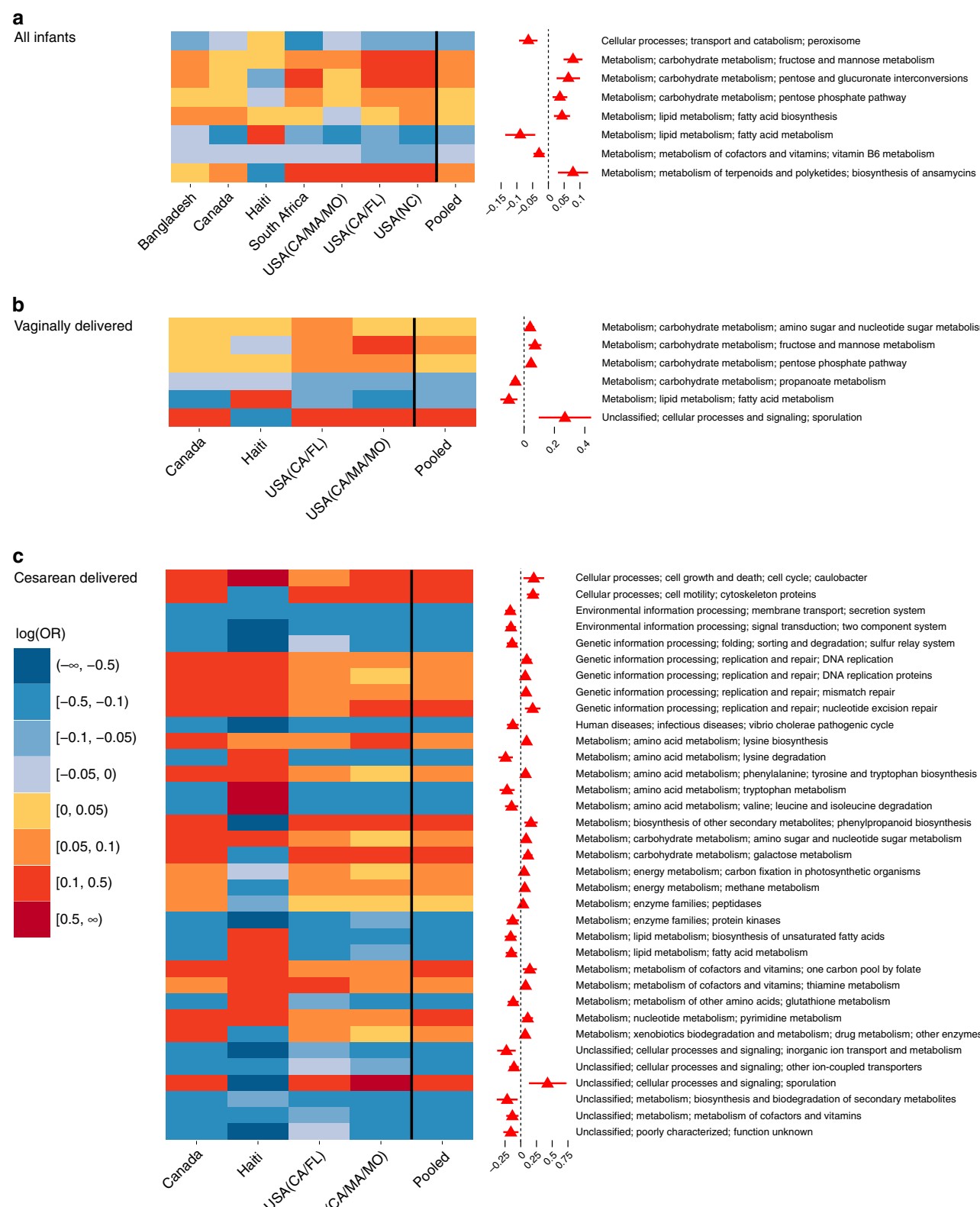

**Duration of EBF and diarrhea-related microbiota dysbiosis.** Again, using data from the Bangladesh study, we found that after adjusting for age of infants at sample collection, diarrhea at the time of sample collection (vs. no diarrhea) was associated with reduced gut microbiota age in infants who received less than 2 months of EBF (MD = −1.17 months, 95% CI = [−2.11;

−0.23], $p = 0.01$; Fig. 5c). Diarrhea in infants who received less than 2 months of EBF was also associated with reduced gut microbial diversity, as showed by Shannon index (DD = −0.58, 95% CI = [−0.83, −0.34], $p < 0.0001$; Fig. 5d) and the three other common alpha diversity indices (Supplementary Fig. 18). In contrast, no diarrhea-associated differences in gut microbiota age

**Fig. 4** Effects of non-EBF vs. EBF on gut bacterial pathway abundances in infants ≤6 months of age. **a** Meta-analysis of all infants in all seven included studies: heatmap of log(odds ratio) (log[OR]) of relative abundances of gut microbial KEGG pathways at level 3 between non-EBF and EBF infants for each study and forest plot of pooled estimates of all seven studies with 95% confidence intervals (95% CI). **b** Meta-analysis of vaginally born infants in four studies: heatmap of log(OR) of relative abundances of gut microbial KEGG pathways at level 3 between non-EBF and EBF infants for each study and forest plot of pooled estimates of four studies with 95% CI. Only four studies with available birth mode information (Canada, Haiti, USA [CA–MA–MO] and USA [CA–FL]) are included. **c** Meta-analysis of C-section born infants in four studies: heatmap of log(OR) of relative abundances of gut microbial KEGG pathways at level 3 between non-EBF and EBF infants for each study and forest plot of pooled estimates of four studies with 95% CI. Only four studies with available birth mode information (Canada, Haiti, USA (CA–MA–MO) and USA (CA–FL)) are included. All log(OR) estimates of each pathway from each study were from generalized additive models for location scale and shape (GAMLSS) with zero-inflated beta family (BEZI) and were adjusted for age of infants at sample collection. Pooled log(OR) estimates and 95% CI (forest plot) were from random-effects meta-analysis models based on the adjusted log (OR) estimates and corresponding standard errors of all included studies. Pooled log(OR) estimates with pooled p-values < 0.05 are in red and those with false discovery rate (FDR)-adjusted pooled p-values < 0.1 are shown as triangles. Only pathways with FDR-adjusted pooled p-value < 0.1 are shown. EBF exclusive breastfeeding, non-EBF non-exclusive breastfeeding, KEGG Kyoto Encyclopedia of Genes and Genomes, OR odds ratio, USA United States of America, CA California, FL Florida, MA Massachusetts, MO Missouri, NC North Carolina

or microbial diversity were observed in infants who received more than 2 months of EBF (all p-values for heterogeneity tests < 0.05).

Diarrhea at the time of sample collection was also associated with major perturbation in the gut bacterial composition of infants who received less than 2 months of EBF with a significant increase in the relative abundance of family Streptococcaceae and a significant decrease in the relative abundances of Bifidobacteriaceae and Coriobacteriaceae (all p-values < 0.05 and FDR-adjusted p-values < 0.1). These changes in microbial composition were not observed in infants who received more than 2 months of EBF (Fig. 5e, Supplementary Table 12). Diarrhea at the time of sample collection was associated with an even more striking perturbation in the gut bacterial composition in infants who were not concurrently being breastfed, with a large outgrowth of Streptococcaceae and a tremendous decrease in the relative abundance of Bifidobacericeae (all p-values < 0.05). These perturbations were almost absent in infants receiving breast milk at the time of diarrhea (Fig. 5f, Supplementary Table 12). The incidence of diarrhea was not different between breastfeeding statuses.

## Discussion

Our study analyzed data from seven microbiome studies and performed meta-analysis pooling estimates across studies with a total of 1825 stool samples of 684 infants from five countries. We found remarkably consistent differences between non-EBF and EBF infants in gut microbial diversity, microbiota age, microbial composition, and microbial predicted functional pathways. The infants' mode of delivery was associated with modification of these differences. We also found notable interaction effects between breastfeeding and diarrhea on infant gut microbiota differences. With large datasets combined from different populations, our results are more robust and generalizable than from a single study.

Prior studies have reported increased bacterial species richness or diversity in non-EBF vs. EBF[27] and/or trends of increased bacterial diversity across EBF, non-EBF, and non-BF[28,30,31]. However, studies report different indices, analyze data in different ways, and some do not account for age of infants at the time of stool sample collection, which is associated with breastfeeding status and infant gut microbiota. Our results showed a significant and consistent increase in all four commonly used alpha diversity indices in non-EBF vs. EBF infants ≤6 months of age after adjusting for age of infants at sample collection. We also showed a consistent increase in gut microbiota age in non-EBF infants vs. EBF infants before 6 months of age across studies. We speculate that a more stable, less diverse gut microbiota, associated with EBF, may be necessary in the early months of development.

There was substantial heterogeneity across studies and populations in gut bacterial taxonomic composition and gut bacterial metabolic pathway composition differences between non-EBF and EBF infants in the first 6 months of life. For example, the decrease in relative abundance of Proteobacteria in non-EBF vs. EBF infants was observed in four studies in North America, but the opposite was observed in other studies in Bangladesh, Haiti, and South Africa. This heterogeneity may be due to dietary differences or differences in formula ingredients in the non-EBF group across different populations. In addition, the gut microbiota of EBF infants, which is largely seeded by their mothers' breastmilk microbiota and HMOs[3,5,9,12], might also be influenced by the mother's diet or other exposures, which might also be different across populations[10]. Infant ethnicity has been reported to influence the infant gut microbiota[35]. In addition, variation in the region of 16S RNA gene targeted between studies may also contribute to heterogeneity. Despite this expected variation, our results revealed some important consistencies across populations. Our results showed a consistent increase in the relative abundance of Bacteroidetes in non-EBF vs. EBF infants in all seven studies, as well as an overall significant increase in relative abundance of Firmicutes. More specifically, our results showed a persistent increase in relative abundances of genera Bacteroides, Eubacterium, and Veillonella in non-EBF vs. EBF infants. While Bifidobacterium is the most common bacterial genus in gastrointestinal tract of young infants, Bacteroides and Eubacterium are the most common bacterial genera in the gastrointestinal tract of adults[36,37]. Though these genera may be part of normal gut bacterial community, the increase in abundance of gut Bacteroides has been shown to be associated with higher body mass index (BMI) in young children[38] and Veillonella can be associated with different types of infection[39]. Our results also showed a consistent increase in relative abundances of major microbial predicted pathways related to carbohydrate metabolism, as well as a consistent decrease in relative abundances of crucial pathways related to lipid metabolism/homeostasis, free radical detoxification, and metabolism of cofactors and vitamins in non-EBF infants[40]. These findings may provide insight into biological mechanisms for the higher risk of obesity, diabetes, and other adverse health outcomes in children who were not breastfed or non-exclusively breastfed in early months of life[24–26].

Interestingly, our results revealed notable heterogeneity regarding the perturbation in predicted microbial functional pathways associated with non-EBF stratified by mode of delivery. The remarkably larger number of perturbed pathways of different cellular and metabolic processes in infants of cesarean deliveries may suggest that gut microbiota in these infants are more vulnerable to the effects of non-EBF. We also observed that non-EBF infants had a much lower abundance of Proteobacterial species

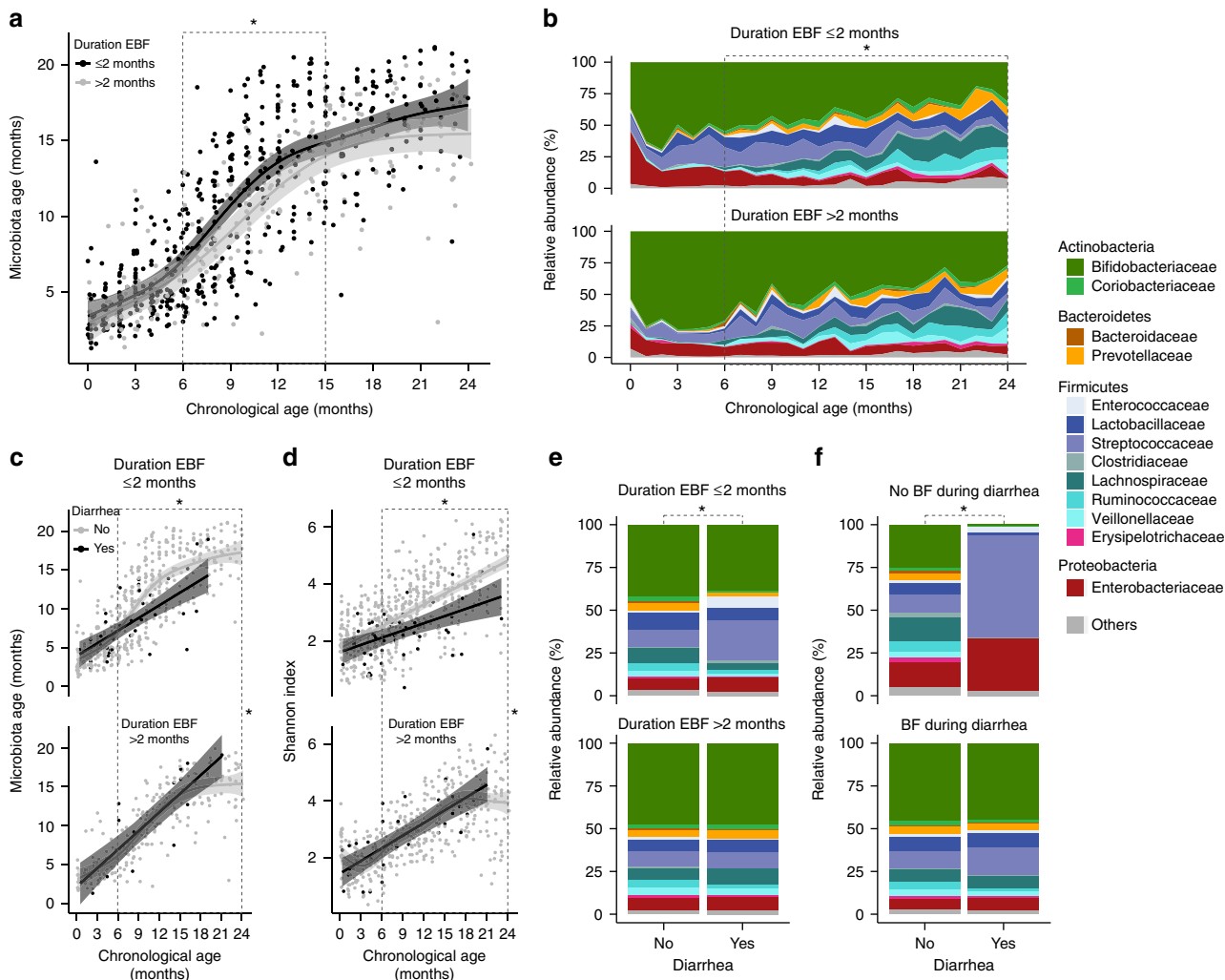

**Fig. 5** The continued effects of EBF on the infant gut microbiota up to 2 years of age. Data from Bangladesh study only. **a** The impact of duration of EBF (shorter than 2 months vs. longer than 2 months from birth) on gut microbiota age. **b** The impact of duration of EBF on gut bacterial family composition. **c** The effects of diarrhea (vs. no diarrhea) around the time of stool sample collection on gut microbiota age in infants with duration of EBF shorter than 2 months vs. longer than 2 months from birth. **d** The effects of diarrhea (vs. no diarrhea) around the time of stool sample collection on gut microbial diversity (Shannon index) in infants with duration of EBF shorter than 2 months vs. longer than 2 months from birth. **e** The effects of diarrhea (vs. no diarrhea) around the time of stool sample collection on gut bacterial taxa composition at the family level in infants with duration of EBF shorter than 2 months vs. longer than 2 months from birth. **f** The effects of diarrhea (vs. no diarrhea) around the time of stool sample collection on gut bacterial taxa composition at the family level in infants receiving no breastfeeding at the time of diarrhea vs. infants receiving breastfeeding at the time of diarrhea. Number of infants $n = 50$ (duration of EBF $\leq 2$ months $n = 30$, duration of EBF > 2 months $n = 20$). Number of samples 0–2 years of age $n_s = 996$ (duration of EBF $\leq 2$ months $n_s = 580$, duration of EBF > 2 months $n_s = 416$). Number of samples 6 months to 2 years of age $n_s = 674$ (duration of EBF $\leq 2$ months $n_s = 378$ [diarrhea $n_s = 29$, no diarrhea $n_s = 349$]; duration of EBF > 2 months $n_s = 296$ [diarrhea $n_s = 19$, no diarrhea $n_s = 277$]; with breastfeeding $n_s = 616$ [diarrhea $n_s = 45$, no diarrhea $n_s = 571$]; without breastfeeding $n_s = 44$ [diarrhea $n_s = 2$, no diarrhea $n_s = 42$]). Fitted lines and 95% confidence intervals (95% CI) were from generalized additive mixed models (GAMM). Gray dashed lines demarcate time periods tested. Black stars indicate statistical significance. EBF exclusive breastfeeding, BF breastfeeding

than EBF infants after cesarean delivery. It appears that when gut microbiota are depleted with Bacteroidetes, as characteristically in early infancy after cesarean delivery, formula feeding further depletes Proteobacteria[41]. These findings may shed light on the mechanisms for the higher risk of adverse health outcomes in infants delivered by cesarean section[42–44] and emphasize the apparent importance of EBF in cesarean-delivered infants. Differences in breastfeeding practices by mode of delivery may also account for these findings[45].

Differences between EBF and non-EBF infant gut microbiota in the first 6 months of life were still evident between 6 months to 2 years of age. Shorter duration of EBF was associated with

increased gut microbiota age, as well as earlier and larger increases in relative abundances of many bacterial families other than the beneficial family Bifidobacteriaceae. In contrast, longer EBF was associated with a more stable bacterial composition in the early months of life and higher relative abundance of Bifidobacteriaceae. These findings again support the hypothesis that early changes in gut microbiota associated with non-EBF may be disproportional to immunological and biological maturity of infants in early months. That is, EBF nourishes a stable gut bacterial taxa composition that may be beneficial for the infants who are still immature in early months of life. Our results are also consistent with the published literature that exposures in early

**Table 1 Summary of the included studies**

| Data origin, study population (reference) | Study design, sample size (≤6 months of age) | Breastfeeding categories, definition, and number of samples (n) | Region of 16S rRNA genes/sequence platform |
|---|---|---|---|
| Bangladesh (Subramanian et al. 2014)[34] [a, c, d] | Longitudinal monthly stool sample collection during the first 2 years after birth of 50 healthy Bangladeshi children (25 singletons, 11 twin pairs, and one set of triplets). Number of samples ≤6 months of age: 322 | Three categories: (1) EBF: fed breast milk without formula or solid food ($n = 138$); (2) Non-EBF: fed breast milk plus either formula or solid ($n = 178$); (3) Non-BF: fed formula or solid food without breast milk ($n = 6$) | V4 /Illumina MiSeq |
| Canada (Azad et al. 2015)[30] [a, b] | One-time sample collection of 167 infants around 3 months of life (a subset of the Canadian Healthy Infant Longitudinal Development (CHILD) national population-based birth cohort). Number of samples ≤ 6 months of age: 167 | Three categories: (1) EBF: fed breast milk without formula or solid food ($n = 86$ (vaginally born = 68, C-section born = 18)); (2) non-EBF: fed breast milk plus either formula or solid ($n = 48$ (vaginally born = 36, C-section born = 12)); (3) non-BF: fed formula or solid food without breast milk ($n = 33$ (vaginally born = 26, C-section born = 7)) | V4/Illumina MiSeq |
| Haiti (Bender et al. 2016)[3] [b, c] | One-time stool sample collection of 48 HIV-negative infants with age varied from 0 to 6 months whose mothers were HIV negative ($n = 25$) or HIV positive ($n = 23$). Number of samples ≤ 6 months of age: 48 | Two categories: (1) exclusive breastfeeding (EBF): fed only breast milk ($n = 37$ (vaginally born = 32, C-section born = 5)); (2) non-EBF: fed breast milk plus anything other than breast milk ($n = 11$ (vaginally born = 10, C-section born = 1)) | V4 /Illumina MiSeq |
| South Africa (Wood et al. 2018)[27] | Longitudinal stool sample collection of 72 healthy infants of HIV-negative mothers at birth, 6, and 14 weeks. Number of samples ≤ 6 months of age: 143 | Two categories: (1) EBF: fed breast milk exclusively, except for prescribed medicine ($n = 86$); (2) non-EBF: fed breast milk plus any other foods, including traditional medicines and water ($n = 57$) | V4 /Illumina MiSeq |
| USA (California and Florida) (Pannaraj et al. 2017)[5] [a, b, c] | Longitudinal stool sample collection of 113 healthy full-term infants at 0–7 days, 8–30 days, 31–90 days, and 91–180 days. Number of samples ≤ 6 months of age: 230 | Three categories: (1) EBF: fed only breast milk ($n = 150$ (vaginally born = 99, C-section born = 48)); (2) non-EBF: fed breast milk plus either formula or solid ($n = 68$ (vaginally born = 53, C-section born = 15)); (3) Non-BF: fed formula and/or solid without breast milk ($n = 12$ (vaginally born = 9, C-section born = 2)); nine samples with unknown feeding category; four samples with unknown birth mode | V4 /Illumina MiSeq |
| USA (Massachusetts, Missouri, and California) (Sordillo et al. 2017)[18] [a, b] | One-time stool sample collection of 228 infants at age 3 to 6 months who were enrolled in Vitamin D Antenatal Asthma Reduction Trial (VDAART), a clinical trial of vitamin D supplementation in pregnancy to prevent asthma and allergies in offspring. Number of samples ≤ 6 months of age: 220 | Three categories: (1) EBF: fed breast milk without formula or solid food ($n = 38$ (vaginally born = 28, C-section born = 10)); (2) Non-EBF: fed breast milk plus either formula or solid ($n = 66$ (vaginally born = 45, C-section born = 21)); (3) non-BF: fed formula or solid food without breast milk ($n = 116$ (vaginally born = 72, C-section born = 44)); eight samples with unknown feeding category | V3–V5/pyrosequencing (Roche 454 Titanium) |
| USA (North Carolina) (Thompson et al. 2015)[28] [a, c] | Longitudinal stool sample collection of six healthy full-term infants with varied age. Number of samples ≤ 6 months of age: 21 | Three categories: (1) EBF: fed breastmilk without formula or solid food ($n = 12$); (2) Non-EBF: fed breastmilk plus either formula or solid ($n = 8$); (3) Non-BF: fed formula or solid food without breast milk ($n = 1$) | V1–2/Roche GS FLX Titanium |

[a]Studies with three breastfeeding categories (exclusive breastfeeding (EBF), non-exclusive breastfeeding (non-EBF), non-breastfeeding (non-BF)) used for trend tests across three categories
[b]Studies with available birth mode information used for meta-analysis stratified by birth mode
[c]Studies with available infant sex information used for the analyses adjusting for infant age and sex
[d]This study contains data from 6 months to 2 years of age, which were used for the analysis from 6 months to 2 years of age. Data from this study were downloaded from the authors' website: https://gordonlab.wustl.edu/Subramanian_6_14/Nature_2014_Processed_16S_rRNA_datasets.html. Data from six other studies were obtained directly from the investigators.
Additional summaries of these included studies are in Supplementary Table 13

life, such as duration of EBF can affect the establishment of the gut microbiota in older children and adults[35,46–49] and may help explain the mechanism for the short-term and long-term health effects of EBF in early months of life.

Another particularly intriguing finding from our analysis is the apparently protective effect of EBF on the infant gut microbiota during diarrheal episodes. Diarrhea has been previously shown to cause perturbations in the gut microbiota[34,50]. Our analysis revealed that diarrhea was associated with a loss of microbial diversity, microbiota age, and the relative abundance of Bifidobacteriaceae, as well as an increase in the relative abundance of Streptococcaceae. Remarkably, these changes were almost completely absent in infants who received more than 2 months of EBF, as well as in those who were being breastfed at the time of diarrhea. These findings support the apparent importance of longer duration of EBF in the first 6 months of life and

continuation of breastfeeding after 6 months of life in maintaining a homeostatic gut microbiota that may be more resistant to outgrowth of pathogenic microbes that lead to diarrhea. Taken together, our findings support a role for gut commensal bacteria in mediating protective effects of breastfeeding on diarrhea morbidity and mortality.

In terms of methodology, our study applied robust methodological approaches for the analysis of microbiome data and meta-analysis across microbiome studies. Standardization of alpha diversity indices and predicted microbiota age from each study makes the estimates of these measures comparable between studies. Zero-inflated beta GAMLSS models allow proper examination of relative abundances of bacterial taxa and predicted functional pathways, which range from zero to one, and are generally zero-inflated as well as adjustment for confounding covariates and handling longitudinal or cross-sectional data. The estimates from zero-inflated beta GAMLSS models are the difference in log odds of relative abundances between groups and thus are comparable between studies. All effect estimates in our analyses were adjusted for variation in age of the infant at sample collection, which might largely influence the infant gut microbiota composition as well as breastfeeding status but was not routinely accounted for in the analysis[3,28] or was partially accounted by study design (collecting samples at similar infant age)[30,32] in some published studies. There have been some published meta-analyses for microbiome data[51–56], but none of these addressed between-group comparison pooled effects when combining data from many studies as done here. The use of random-effects meta-analysis models pooling estimates from studies allows examination of study-specific effects, the heterogeneity between studies, and the overall pooled effects across studies. In addition, although generalized additive (mixed) models (GAMs/GAMMs) may be prone to overfitting as compared to generalized linear (mixed) models (GLMs/GLMMs), the use of GAMs/GAMMs in our study allows flexibility in examining any linear/non-linear relationship and difference in curves between the groups. This is especially useful when there may be a difference between multiple groups within a study and across multiple studies. Nevertheless, careful checking of model fits is necessary, and our plots for GAMMs model fits and observed data (e.g. Fig. 5a, c, d) do not show obvious overfitting.

This study has some limitations. First, definitions of EBF and non-EBF were not identical across the seven included studies. Specifically, for the five studies in Bangladesh, Canada, CA–FL, North Carolina, and CA–MA–MO, EBF was defined as ingestion of only breastmilk without formula or solid food and non-EBF as ingestion of breastmilk plus formula and/or solid food. By contrast, the two studies in Haiti and South Africa followed World Health Organization (WHO) guidelines, with EBF as feeding only breastmilk and non-EBF as feeding breast milk plus anything other than breast milk including traditional medicines or water. This difference might contribute to some of the variation in results across studies. Second, several of the studies were of a relatively small size or included other potential issues. Specifically, the Haiti study included only 48 infants, of which half were born to HIV-infected mothers, which has been shown to influence the infant gut microbiota[3]. The North Carolina study included a very small number of samples in infants ≤6 months old (n = 21). The VDAART trial (CA–MA–MO) study included samples from infants who were at high risk for asthma and allergies, half of whom were randomized to vitamin D supplementation in pregnancy. However, sensitivity meta-analyses excluding these studies showed similar findings to the overall meta-analysis, suggesting that our results are robust. Finally, all of the results pertaining to samples collected after 6 months of age were from a single study (Bangladesh) and thus should be replicated in other cohorts.

However, this study included nearly 1000 samples collected monthly from birth to 2 years of life in 50 subjects with detailed meta-data[34].

In conclusion, our meta-analysis revealed consistent findings across populations that may help elucidate the effects of EBF on the infant gut microbiota. Non-EBF or shorter duration of EBF in the first 6 months of life was associated with higher gut microbial diversity, higher microbiota age, bacterial composition more closely resembling the adult microbiota, higher relative abundance of bacterial functional pathways related to carbohydrate metabolism, and lower relative abundance of bacterial functional pathways related to lipid metabolism, detoxification, and cofactor and vitamin metabolism. The differences in microbial functional pathways associated with non-EBF were larger in cesarean-delivered infants than in vaginally delivered infants. Furthermore, EBF, especially longer than 2 months from birth, was associated with a more stable gut bacterial taxa composition and reduced diarrhea-associated microbial dysbiosis. The early and large change associated with non-EBF in the infant gut microbiota may be disproportional to age-appropriate immunological and biological development of the infant. Altogether, our results support a consistent role of EBF to maintain a homeostatic developmental trajectory of the infant gut microbiota and shed light on the mechanisms of the short-term and long-term benefits of EBF in the first 6 months of life.

## Methods

**Data sources and study population**. Processed and partially processed 16S rRNA gene sequence data of stool samples were obtained from seven previously published studies[3,5,18,27,28,30,34]. The reuse of these published data for our meta-analysis complies with all relevant ethical regulations. Of the included studies, three were from the US, one from Canada, one from Haiti, one from South Africa, and one from Bangladesh. There were five studies with three breastfeeding categories (EBF, non-EBF, and non-BF) and two studies with two breastfeeding categories (EBF and non-EBF). The total number of samples of infants ≤6 months of age included in the overall meta-analyses was 1151 (EBF n = 547, non-EBF n = 436, non-BF n = 168). There were four studies with available information regarding the infants' mode of delivery included in meta-analyses stratified by mode of delivery with a total number of samples of 670 (vaginal deliveries n = 484, cesarean deliveries n = 186). The Bangladesh study contained 674 samples from 6 months to 2 years of age. In total, 1825 samples of 684 infants were used in our analyses. A summary of the included studies, data characteristics, and prior data processing are presented in Table 1 and Supplementary Table 13.

**Data processing**. Sequence data from each included study were processed separately. To achieve necessary data consistency for meta-analyses in this study, OTU picking was performed at 97% similarity using QIIME version 1.9.1[57], with the Greengenes database (version 13.8)[58]. Alpha rarefaction was done in QIIME using default options. Rarefaction depth was selected as the highest depth that retained all study samples. Taxonomic relative abundances from phylum to genus levels and alpha diversity indices were calculated based on rarefied OTU tables. Metagenomic functional compositions of stool bacterial communities were predicted based on the normalized OTU tables using PICRUSt[59], and relative abundances were then calculated for the resulting Kyoto Encyclopedia of Genes and Genomes (KEGG) functional pathways[40].

**Statistical analysis**. For each study, mean alpha diversity indices were calculated for each sample at the selected rarefaction depth. RF modeling of gut microbiota maturity has been widely used to characterize development of the microbiota over chronological time[5,34,48]. Adapting the approach from Subramanian et al.[34], relative abundances of 36 bacterial genera (Supplementary Table 1) that were detected in the data of all seven included studies were regressed against infant chronological age using a RF model on the training dataset of the Bangladesh study. The RF model fit based on relative abundances of these shared bacterial genera was then used to predict infant age on the test data of the Bangladesh study and the data of each other included study. The predicted infant age based on relative abundances of these shared bacterial genera in each study is referred to as gut microbiota age in this paper. Alpha diversity indices and microbiota age from each study were standardized to have a mean of 0 and standard deviation of 1 to make these measures comparable across studies. Generalized additive mixed model (GAMM) as well as linear mixed-effect model with subject random intercept (for longitudinal data) or linear model (for non-longitudinal data) adjusted for infant age at the time of stool sample collection were used to further examine the curves of standardized

alpha diversity indices and standardized gut microbiota age over age of infants, as well as the difference between the groups in each study.

For each study, the summary tables of bacterial taxa and pathway relative abundances were filtered to retain only the taxa and pathways that had an average relative abundance of at least 0.005% and were detected in at least 5% of the number of samples in that study. Relative abundances of bacterial taxa and bacterial KEGG metabolic pathways were examined using generalized additive models for location scale and shape (GAMLSS), with zero-inflated beta family (BEZI) and (mu) logit links and other default options as implemented in the R package gamlss[60]. This approach allows proper examination of microbiome relative abundance data, which range from 0 to 1, and are generally zero-inflated, as well as adjustment for covariates (e.g. infant age at sample collection) and handling of longitudinal data by including a subject random effect. In each study, to roughly test for trends across three breastfeeding categories (EBF, non-EBF, and non-BF), breastfeeding was coded as a continuous variable in the models.

To examine the overall effects while addressing heterogeneity across studies, random-effects meta-analysis models with inverse variance weighting and DerSimonian–Laird estimator for between-study variance were used to pool the adjusted estimates and their standard errors from all included studies. Meta-analyses were done for only bacterial taxa and pathways whose adjusted estimates and standard errors were available in at least 50% of the number of included studies.

All statistical tests were two sided. $p$-values < 0.05 were regarded as significant and false discovery rate (FDR)-adjusted $p$-values < 0.1 were regarded as significant after adjusting for multiple testing. All analyses were done using custom code in R statistical software version 3.4.2[61].

**Code availability**. The R code used to generate the results in this paper is available in Github [https://github.com/nhanhocu/metamicrobiome_breastfeeding] and Zenodo [https://doi.org/10.5281/zenodo.1304367][62].

## Data availability
All data used in the analyses of this study are included in the following published articles and their supplementary information files[3,5,18,27,28,30,34]. The data from the Bangladesh study were downloaded from the authors' website [https://gordonlab.wustl.edu/Sub-ramanian_6_14/Nature_2014_Processed_16s_rRNA_datasets.html]. The data from six other studies were obtained directly from the investigators. The datasets that support the findings of this meta-analysis are available in Github [https://github.com/nhanhocu/metamicrobiome_breastfeeding] and Zenodo [https://doi.org/10.5281/zenodo.1304367][62].

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

## Acknowledgements

We would like to acknowledge the Canadian Healthy Infant Longitudinal Development (CHILD) study investigators for providing the data of the study in Canada. A full list of consortium members appears in Supplementary Note 1. Funding: This work was supported by Mervyn W. Susser fellowship in the Gertrude H. Sergievsky Center, Columbia University Medical Center (to N.T.H.); National Institute of Dental and Craniofacial Research grants R01 DE 0212380 (to L.K.); NIH grant numbers U01HL091528 and R01HL108818 (to S.T.W. and A.A.L.); NICHD grant number K08HD069201, a developmental grant from the University of Washington Center for AIDS Research (CFAR), an NIH funded program under the award number AI027757, and the following NIH Institutes and Centers: NIAID, NCI, NIMH, NIDA, NICHD, NHLBI, NIA, NIGMS, and NIDDK (to H.B.J.); NIH/NIDDK grant number P30 DK34987 (to M.A.A.P.); NIH Grant numbers K23 HD072774-02 (to P.S.P.), K12 HD 052954-09 (to J.M.B.), and UM1AI106716 (to G.M.A.); grant 227312 from the Canadian Institutes of Health Research Canadian Microbiome Initiative (to A.L.K.), the Canadian Institutes of Health Research and the Allergy, Genes, and Environment (AllerGen) Network of Centres of Excellence for the CHILD study.

## Author contributions

N.T.H. and L.K. conceived the study. F.L., K.A.L., H.M.T., B.B., P.S.P., J.M.B., M.B.A., A.L.T., S.T.W., M.A.A.P., A.L.K., A.A.L., H.B.J. and G.M.A. provided data. N.T.H. performed the data analysis with inputs from F.L. and L.K. N.T.H., F.L., H.B.J., G.M.A. and L.K. prepared the manuscript with inputs from all authors. All authors read and approved the final manuscript.

## Additional information

**Competing interests:** The authors declare no competing interests.

