## [Peer Review File · Nature Communications]

Reviewers' comments:

Reviewer #1 (Remarks to the Author):

The authors are to be commended for summarizing a diverse set of results across populations. It does help to know that despite heterogeneity there are some commonalities. This is a nice piece of summary analysis that re-emphasizes a lot that we already knew. The main findings about maturity of the MB and with non exclusive BF and c-section are as expected. Unfortunately it really doesn't shed light toward mechanisms as the authors suggest. The "big picture" is still not changed. The results from Kegg analysis are still conjecture. There are no new hypotheses generated from the data that would help one investigate this field further.

Reviewer #2 (Remarks to the Author):

This study is a meta-analysis on the effects of breastfeeding on infant gut microbiota (focusing on the first 6 months of age), covering 7 studies from different populations, 1825 stool samples, and 684 infants. The study is motivated by the notable variation in the literature regarding the associations between breastfeeding and gut microbiota composition, due to differing study protocols and populations, treatment of potential confounding variables, and the large overall variation in infant gut microbiome.

Based on the meta-analysis, the authors conclude that in breastfed babies, elevated levels are reported in gut microbiota richness, maturity, Bacteroidetes-Firmicutes ratio, and carbohydrate metabolism pathways. In non-breastfed babies, elevated levels are reported in certain metabolic processes. In addition, C-section was associated with higher microbiota perturbations in the non-breastfed group, compared to vaginal delivery. Breastfeeding also mitigated diarrhea-associated dysbiosis. These observations are mostly consistent across different populations and supports the hypothesis that breastfeeding modulates infant gut microbiota for beneficial health effects.

Major comments:

As this a meta-analysis, the results have limited novelty but the paper does good work in summarizing the existing evidence and providing robust conclusions that are supported by multiple studies, which are properly cited in the manuscript. The main contribution, and impact in the field, is in establishing a more robust understanding of previously reported associations, rather than in reporting new associations. The work is of interest in particular for researcher who are studying the effects of gut microbiome on infant development and health. The paper can also provide some guidance on how to carry out meta-analyses in the context of human microbiome research but this interested is limited as overall the meta-analysis is based on previously established methodology.

The work has a suitable length and it is well written with clear English, although I have some minor suggestions for improvement (see below). All claims are properly supported by the data; the authors use appropriately moderate language to report their observations and provide relevant literature references.

The work is also mostly technically sound and as such convincing overall, although I have listed some comments on the details below. Code and data are available in Github. Since this is a commercial platform with no guarantees on long-term storage, it would be advisable that in addition to sharing the digital material via Github, the authors would store the exact version used in this publication in a

suitable academic or institutional repository with a DOI; Data Dryad, Zenodo, Figshare are among popular options but there are many other local, national and international services available. I did not replicate the analyses but the documentation seems clear and appropriate. I also positively acknowledge that the authors have made a preprint already available for the community. Ethical concerns of the study are limited to rights to reuse previously published data. The data sets were obtained either from public sources, or from the original authors, therefore these issues seem to be taken into account. It might be good to cross-check that no confidential data is shared in the supplements and/or data repository.

I have the following major comments on the work and methodology:

- The analyses are based on compositional data (see for instance line 357 and paragraph starting from line 449 but also overall the Methods): CLR-transformed data would be more appropriate for estimating log-odds and other measures of effect size and significance. This should be discussed, and preferably implemented but I wonder if that is possible given the availability of original data from varying sources. For discussion and references on compositionally robust transformations and their relevance, see for instance <https://www.frontiersin.org/articles/10.3389/fmicb.2017.02224/full> It would be also interesting to see whether CLR-transformed abundance data, which should be free of compositionality biases in sample-wise comparison, would lead to improved predictions of infant age.

- The use of GAM models is prone to overfitting compared to GLMMs and hence often not used unless a specific need arises compared to GLMMs (ref: standard literature on generalized linear models). It is not clear whether the use of GAMs is necessary in this study, compared to the simpler GLMMs. Some justification for the use of GAMs over GLMM should be given. Both allow nonlinear effects, count data, and random terms.

- Not sure if I correctly read the Figures. For instance in Fig. 3 we have pooled estimates in the heatmap (last column). These are positive (red) for both Verrucomicrobiaceae and Planococcaceae in the heatmap (left). But in the lineplot on the right side the effect size is negative for Verrucomicrobiaceae and positive for Planococcaceae. I would expect that the lineplot would correspond to the pooled values in the heatmap. But the figure seems to contradict this, and I could not resolve this by reading the figure caption. Some clarification is needed.

- The bioinformatics/statistics methods are in general appropriate and properly controlled for variations (multiple diversity indices tested; associations controlled for age, health status, and medication to the extent that is possible, given that the data comes from various sources. However, gender is also an important potential confounder and typically recorded in such studies, yet not controlled here. If gender can be controlled this should be done; if not, an explanation and discussion on this effect could be included.

Minor comments:

- The notation varies for p-values, sometimes it is $p=0.001$ and other times $p<0.001$; be consistent. The $p=0.001$ format is preferred by some recommendations.

- line 265: did the mode of delivery really modify the effects? This is a mechanistic claim, requiring experimental verification. I would propose the statement "was associated with modifications". Check also otherwise for similar claims throughout the paper (I could not find any but just in case I may have overlooked something).

- 277: "these results seem paradoxical" -> which results seem paradoxical exactly: the differences in

diversity between EBF and non-EBF, or their associations with stability?

- lines 278-279: what does it mean "out of step with age-appropriate development"? I have difficulties in interpreting this sentence.

- line 350: leave out the word "appropriate"; I would leave it for the readers to assess appropriateness

- line 357: log(odds ratio) more commonly used, and rather standard term in my experience is log-odds

- Supplementary Figure 3: the correlation in the test set is higher than I would expect based on my own experience, and given the huge variability in infant gut microbiome and differences in development. It is interesting to see this predictive performance. The Bangladesh gut maturity study is appropriately cited.

- The number of digits is sometimes higher than the plausible accuracy of the study, given measurement noise etc. I would suggest rounding the effect sizes and p-values to 3 significant digits (or another appropriate number), for instance. This will also improve readability. In particular in Supplementary Tables but also occasionally in the text.

- Minor writing issues; for instance line 142 spaces missing after % mark; and I think "figure" and "table" should be capitalized when referring to the manuscript figures and tables (see for instance line 204, 229); lines 142-143 why not put all within single parentheses, and separate with semicolon, I think it is not good writing practice to put to distinct parentheses immediately following each other (like)(this) vs. (like; this); same applies to lines 193, 202, 224 and elsewhere; line 224: "(MD) =164" should this be (MD=164)? Space missing after %; similar things apply to line 242; and p-value resolution on line 224 too high compared to the expected real resolution, I would just use $p=0.02$; same for line 240 and elsewhere; line 415 and similar elsewhere: "deliveries =484" should be "deliveries n=484"?

- "Supplementary figure" -> should this be "Supplementary Figure" (figure also capitalized)? Same for the tables?

- References are not in a harmonized format, some have First Author et al., other have multiple authors listed. Should be harmonized.

- Anne Salonen, Katri Korpela, and Willem de Vos have also done notable research on infant gut microbiome. You could consider citing their work where relevant, some suggestions include doi: 10.1001/jamapediatrics.2016.0585; doi: 10.1016/j.psyneuen.2015.01.006; doi.org/10.1128/MMBR.00036-17

Figures:

- avoid green-red color scale due to color blindness issues; blue-red is more commonly used
- Font size in supplementary figures could be slightly increased.
- Consider sharper colors than the ggplot2 default color palette; consider grayscale where colors are not really necessary (Fig 5a and 5c and other similar figures)

- Fig. 5a: this could be more clear if just the fitted lines with error bars were shown; not sure if showing the individual trajectories is necessary to show

- Supplementary Table 1: is it possible to normalize the numbers so that they are more interpretable, for instance relative importance in percents (0-100%)?
- Supplementary Table 2: bacterial taxa can be listed in this standard format but for a table I would still consider separating the different taxonomic levels each in their own column (Kingdom, Phylum, Genus..)
- Supplementary Tables 6, 9 (check if others): too many parentheses (USA(...)) in the title?
- Supplementary Figures 7-13: the heatmap cell width could be shrinked for improved readability
- Supplementary Figure 14: how about just using gray color scale here, and highlighting the Diarrhea cases with black (others darkgray). This would improve readability and highlight the interesting pattern more visibly. It is also mildly confusing that healthy state is marked with red now, as red color is often associated with health problems or warning signals; and diarrhea (blue) is lost somewhere in the middle of the figure; increased transparency (alpha) could be used for the error intervals, and the error intervals could be shaded with the same color than their corresponding lines. Now both cases have dark gray variation intervals, which slows down figure readability (or course if you switch to gray scale then this will remain so but even then it would be possible to consider different levels of shading to highlight the difference)

RESPONSE TO REVIEWS

Reviewer #1 (Remarks to the Author):

The authors are to be commended for summarizing a diverse set of results across populations. It does help to know that despite heterogeneity there are some commonalities. This is a nice piece of summary analysis that re-emphasizes a lot that we already knew. The main findings about maturity of the MB and with non exclusive BF and c-section are as expected. Unfortunately it really doesnt shed light toward mechanisms as the authors suggest. The "big picture" is still not changed. The results from Kegg analysis are still conjecture. There are no new hypotheses generated from the data that would help one investigate this field further.

Response: We thank the reviewer for comments but disagree that no new hypotheses are generated to help move the field forward. In our analysis we synthesize data across multiple studies to provide a robust and comprehensive view of the “big picture”. Current published literature regarding the role of exclusive breastfeeding (EBF) vs. non-EBF on the infant gut microbiota, their functionality and potential interaction with other factors is still limited with substantial variation in reported results. We believe our study provides useful and robust findings on different aspects of the gut microbiome including microbial diversity, microbiota age, and composition, all of which have been the subject of varying reports in previous studies. We acknowledge the known limitations of metagenomics analyses with PICRUSt. However, we also provide consistent results regarding microbial predicted functions (KEGG analysis). Our KEGG results regarding the difference between non-exclusive breastfeeding vs. exclusive breastfeeding in bacterial predicted function are novel, biologically sound, and supported by data from multiple studies. Therefore, we believe that these results are more robust and generalizable than those from a single study. In addition, despite using data from only one (albeit large) study, the findings regarding the protective effect of longer duration of exclusive breastfeeding on the infant gut microbiota in case of diarrhea are also novel.

Reviewer #2 (Remarks to the Author):

This study is a meta-analysis on the effects of breastfeeding on infant gut microbiota (focusing on the first 6 months of age), covering 7 studies from different populations, 1825 stool samples, and 684 infants. The study is motivated by the notable variation in the literature regarding the associations between breastfeeding and gut microbiota composition, due to differing study protocols and populations, treatment of potential confounding variables, and the large overall variation in infant gut microbiome.

Based on the meta-analysis, the authors conclude that in breastfed babies, elevated levels are reported in gut microbiota richness, maturity, Bacteroidetes-Firmicutes ratio, and carbohydrate metabolism pathways. In non-breastfed babies, elevated levels are reported in certain metabolic processes. In addition, C-section was associated with higher microbiota perturbations in the non-breastfed group, compared to vaginal delivery. Breastfeeding also mitigated diarrhea-associated dysbiosis. These observations are mostly consistent across different populations and supports the hypothesis that breastfeeding modulates infant gut microbiota for beneficial health effects.

Major comments:

As this a meta-analysis, the results have limited novelty but the paper does good work in summarizing the existing evidence and providing robust conclusions that are supported by multiple studies, which are properly cited in the manuscript. The main contribution, and impact in the field, is in establishing a more robust understanding of previously reported associations, rather than in reporting new associations. The work is of interest in particular for researcher who are studying the effects of gut microbiome on infant development and health. The paper can also provide some guidance on how to carry out meta-analyses in the context of human microbiome research but this interested is limited as overall the meta-analysis is based on previously established methodology.

The work has a suitable length and it is well written with clear English, although I have some minor suggestions for improvement (see below). All claims are properly supported by the data; the authors use appropriately moderate language to report their observations and provide relevant literature references.

The work is also mostly technically sound and as such convincing overall, although I have listed some comments on the details below. Code and data are available in Github. Since this is a commercial platform with no guarantees on long-term storage, it would be advisable that in addition to sharing the digital material via Github, the authors would store the exact version used in this publication in a suitable academic or institutional repository with a DOI; Data Dryad, Zenodo, Figshare are among popular options but there are many other local, national and international services available. I did not replicate the analyses but the documentation seems clear and appropriate. I also positively acknowledge that the authors have made a preprint already available for the community. Ethical concerns of the study are limited to rights to reuse previously published data. The data sets were obtained either from public sources, or from the original authors, therefore these issues seem to be taken into account. It might be good to cross-check that no confidential data is shared in the supplements and/or data repository.

Response: We thank the reviewer for the positive and encouraging comments. Based on the constructive comments/suggestions (below) of the reviewer, our manuscript has been improved substantially. As suggested, we have transferred the Github repository for this project to Zenodo with a DOI: <https://doi.org/10.5281/zenodo.1304367>. We also confirm that no confidential data are shared in any of the data files provided.

I have the following major comments on the work and methodology:

- The analyses are based on compositional data (see for instance line 357 and paragraph starting from line 449 but also overall the Methods): CLR-transformed data would be more appropriate for estimating log-odds and other measures of effect size and significance. This should be discussed, and preferably implemented but I wonder if that is possible given the availability of

original data from varying sources. For discussion and references on compositionally robust transformations and their relevance, see for instance <https://www.frontiersin.org/articles/10.3389/fmicb.2017.02224/full> It would be also interesting to see whether CLR-transformed abundance data, which should be free of compositionality biases in sample-wise comparison, would lead to improved predictions of infant age.

Response: We agree that the CLR transformation is a great approach to deal with constant-sum constraint (CSC) of compositional data. However, we did not use CLR transformation in this microbiome meta-analysis project due to two main reasons:

1) Microbiome compositional data is largely zero-inflated. Before CLR transformation, zero values need to be replaced and accordingly non-zero values need to be adjusted. Zero-replacement (or rounding-error replacement) procedures may artificially introduce bias/error to the data as the proportion of zero values in microbiome data is very large (e.g. for Bangladesh data at genus level, 89% are zero values). Although GAMLSS-BEZI does not deal with CSC issue of compositional data (which is inarguably a limitation), it can address proportional data, the inflation of zero values and thereby avoid the bias of vast data imputation.

2) While CLR transformation may be appealing when being applied to one specific microbiome study, it may be less appealing when being applied to multiple studies for meta-analysis. CLR transformation depends on the geometric mean of abundance values, which in turn depends on the number of identified/usable taxa in each study. As the reviewer anticipated, an important issue that limits the use of CLR transformation in this meta-analysis project is the large difference between data sources (e.g. difference in region of 16S RNA genes sequenced, sequencing techniques, preprocessing, pre-filtering procedures used, types of initial data files shared by the collaborative groups). This leads to large differences in the obtained microbiome data between studies (e.g. large difference in number of identified/usable bacterial taxa between studies). As such, we decided against using the CLR transformation in this meta-analysis as we could not reasonably rule out the possibility of additional biases being introduced.

Nevertheless, as suggested, we implemented different zero-replacement approaches and CLR transformation as options in our R functions for bacterial taxa relative abundance comparison. Moreover, we applied multiplicative Kaplan-Meier smoothing spline (KMSS) replacement of zero values (R package “zCompositions”) followed by CLR transformation (R package “compositions”) for the Bangladesh data. We then performed microbiota age prediction on CLR-transformed abundance at genus level using Random Forest models similar to what we did previously for non-transformed relative abundance of genera. The prediction performance using CLR-transformed data on the training set remains equally good ($R^2=0.95$ vs. 0.95), while the prediction on the test set was worse as compared to the prediction using non-transformed compositional data ($R^2=0.59$ vs. 0.65). We also tried different approaches to deal with zero values before CLR transformation but microbiota age prediction performance on the test set remained <0.6 . Although we did not feel that these results were sufficiently compelling to include in this meta-analysis, we do hope that other users may find the additional options for CLR transformation that we implemented to be of utility.

Performance of age prediction based on genus level relative abundance without transformation (our current result)

Performance of age prediction based on genus level relative abundance after zero replacement and CLR transformation

- The use of GAM models is prone to overfitting compared to GLMMs and hence often not used unless a specific need arises compared to GLMMs (ref: standard literature on generalized linear models). It is not clear whether the use of GAMs is necessary in this study, compared to the simpler GLMMs. Some justification for the use of GAMs over GLMM should be given. Both allow nonlinear effects, count data, and random terms.

Response: We agree with the reviewer that GAM models are prone to overfitting. Indeed, for alpha diversity and microbiota age, it is true that based on the results from GAMs/GAMMs, which do not show a complicated non-linear relationship especially in the analyses using Bangladesh data, equivalent GLMMs can be used. However, we used GAMs/GAMMs primarily for their flexibility in examining any linear/non-linear relationship and difference in curves (e.g. curve may look linear for one group while may look cubic for another group within a study). With GAMs/GAMMs, one does not have to pre-examine the data and specify terms as for GLMMs (e.g. quadratic, cubic term, etc). This is particularly useful when there may be difference between multiple groups within a study and difference across multiple studies. In addition, our plots for GAMs/GAMMs model fits and observed data (e.g. Figure 5a, 5c, 5d) do not show obvious overfitting. We have added justification for our use of GAMs over GLMMs in the “Discussion” regarding methodology.

- Not sure if I correctly read the Figures. For instance in Fig. 3 we have pooled estimates in the heatmap (last column). These are positive (red) for both Verrucomicrobiaceae and Planococcaceae in the heatmap (left). But in the lineplot on the right side the effect size is negative for Verrucomicrobiaceae and positive for Planococcaceae. I would expected that the lineplot would correspond to the pooled values in the heatmap. But the figure seems to contradict this, and I could not resolve this by reading the figure caption. Some clarification is needed.

Response: The reviewer is correct in expecting that the effect size (forest plot) should match the pooled values in the heatmap. Both Verrucomicrobiaceae and Planococcaceae have positive pooled values on both the heatmap and forest plot representation. The vertical dashed line in the middle of the forest plot denotes zero values and separates the positive values on the right from the negative values on the left of the forest plot.

- The bioinformatics/statistics methods are in general appropriate and properly controlled for variations (multiple diversity indices tested; associations controlled for age, health status, and medication to the extent that is possible, given that the data comes from various sources. However, gender is also an important potential confounder and typically recorded in such studies, yet not controlled here. If gender can be controlled this should be done; if not, an explanation and discussion on this effect could be included.

Response: We agree with the reviewer that infant sex should be adjusted. Unfortunately, we only have infant sex information from 4 out of 7 included studies. We performed the analysis adjusting for both infant age and sex and compared the results with the analysis adjusting for infant age using the data of these four studies. We added the results to manuscript text (four result sub-sections regarding alpha diversity, microbiota age, bacterial taxa composition and KEGG pathways) and supplementary materials (Supplementary Figure 3, 7, 15, 17).

Minor comments:

- The notation varies for p-values, sometimes it is $p=0.001$ and other times $p<0.001$; be consistent. The $p=0.001$ format is preferred by some recommendations.

Response: We reformatted p-values to 2 or 3 digits where appropriate and used $p<0.001$ for p-values smaller than 0.001 in the main text as suggested.

- line 265: did the mode of delivery really modify the effects? This is a mechanistic claim, requiring experimental verification. I would propose the statement "was associated with modifications". Check also otherwise for similar claims throughout the paper (I could not find any but just in case I may have overlooked something).

Response: We thank the reviewer for this suggestion and have corrected our wording as suggested.

- 277: "these results seem paradoxical" -> which results seem paradoxical exactly: the differences in diversity between EBF and non-EBF, or their associations with stability?

- lines 278-279: what does it mean "out of step with age-appropriate development"? I have difficulties in interpreting this sentence.

Response: Our intent was to suggest that the increased diversity and microbiota age in non-EBF infants was a surprising result given the typical association between diversity and a "healthy" microbiome. However, we believe that equation of diversity with health is perhaps an oversimplification of the developmental trajectory of the infant gut microbiome. We have removed the confusing sentences and simply state: "We speculate that a more stable, less diverse gut microbiota, associated with EBF, may be necessary in the early months of development"

- line 350: leave out the word "appropriate"; I would leave it for the readers to assess appropriateness

Response: We agree with the reviewer and have removed the word "appropriate".

- line 357: $\log(\text{odds ratio})$ more commonly used, and rather standard term in my experience is $\log\text{-odds}$

Response: the coefficient from the model is the difference in log odds between groups or the log of odds ratio between groups. We have adjusted the term to "difference in log odds" as suggested by the reviewer.

- Supplementary Figure 3: the correlation in the test set is higher than I would expect based on my own experience, and given the huge variability in infant gut microbiome and differences in development. It is interesting to see this predictive performance. The Bangladesh gut maturity study is appropriately cited.

Response: We tried different methods for microbiota age prediction using Random Forest model with the Bangladesh data. The best overall performance was achieved using genus level relative

abundances, so we used this approach. Given the reviewer's experience with these type of data and analyses, we double-checked our predictive models for the Bangladesh data and did not find any errors in the code or calculations. We hope that making all of our analyses available via Github and Zenodo will enable other researchers to confirm and extend our findings.

- The number of digits is sometimes higher than the plausible accuracy of the study, given measurement noise etc. I would suggest rounding the effect sizes and p-values to 3 significant digits (or another appropriate number), for instance. This will also improve readability. In particular in Supplementary Tables but also occasionally in the text.

Response: We rounded all effect size and 95% CI to 2 digits and all p-values to 4 digits for supplementary tables and displayed <0.0001 for p-values smaller than 0.0001 (due to multiple testing, it may be reasonable to display 4 digits or more for p-values in Supplementary Tables). We also reformatted the number of digits in the manuscript text where appropriate as per the comment above.

- Minor writing issues; for instance line 142 spaces missing after % mark; and I think "figure" and "table" should be capitalized when referring to the manuscript figures and tables (see for instance line 204, 229); lines 142-143 why not put all within single parentheses, and separate with semicolon, I think it is not good writing practice to put to distinct parentheses immediately following each other (like)(this) vs. (like; this); same applies to lines 193, 202, 224 and elsewhere; line 224: "(MD) =164" should this be (MD=164)? Space missing after %; similar things apply to line 242; and p-value resolution on line 224 too high compared to the expected real resolution, I would just use $p=0.02$; same for line 240 and elsewhere; line 415 and similar elsewhere: "deliveries =484" should be "deliveries n=484"?

- "Supplementary figure" -> should this be "Supplementary Figure" (figure also capitalized)? Same for the tables?

Response: We added space after % marks, capitalized all "Figure" and "Table", added n before number of samples, combined parentheses, etc.

- References are not in a harmonized format, some have First Author et al., other have multiple authors listed. Should be harmonized.

Response: The reviewer is correct that the format of references seems inconsistent. However, after checking and comparing with published articles in Nature Communications, we find that this is the reference style for Nature journals (list author 1 et al for the paper with many authors; list several authors for the paper with few authors).

- Anne Salonen, Katri Korpela, and Willem de Vos have also done notable research on infant gut microbiome. You could consider citing their work where relevant, some suggestions include doi:10.1001/jamapediatrics.2016.0585; doi:10.1016/j.psyneuen.2015.01.006; doi.org/10.1128/MMBR.00036-17

Response: We thank the reviewer for the informative and interesting references. We have added the suggested references in our revised manuscript text (references 13 and 50). We did not include reference doi:10.1016/j.psyneuen.2015.01.006 as it is about the association between prenatal stress and infant intestinal microbiota which is less relevant to our paper.

Figures:

- avoid green-red color scale due to color blindness issues; blue-red is more commonly used

Response: We changed relevant figures to a blue-red color scale.

- Font size in supplementary figures could be slightly increased.

Response: We increased the font size in supplementary figures.

- Consider sharper colors than the ggplot2 default color palette; consider grayscale where colors are not really necessary (Fig 5a and 5c and other similar figures)

- Fig. 5a: this could be more clear if just the fitted lines with error bars were shown; not sure if showing the individual trajectories is necessary to show

Response: We switched to greyscale for Figure 5a, 5c, 5d. We tried greyscale for Figure 1a and Figure 2a but when there are more than two groups, it is not clearly distinguishable between groups with greyscale. We decided to use sharper colors for Figure 1a and Figure 2a instead. We kept individual trajectories for Figure 1a and Figure 2a to distinguish longitudinal data from one time/cross-sectional data. For Figure 5a, 5c, 5d, because the data was from only the Bangladesh study which was already described as longitudinal, we removed individual trajectories to make it clearer as suggested.

- Supplementary Table 1: is it possible to normalize the numbers so that they are more interpretable, for instance relative importance in percents (0-100%)?

Response: We added another column for relative importance as a percentage in Supplementary Table 1.

- Supplementary Table 2: bacterial taxa can be listed in this standard format but for a table I would still consider separating the different taxonomic levels each in their own column (Kingdom, Phylum, Genus..)

Response: We reformatted bacterial taxa in all relevant Supplementary Tables as suggested. As an exception for Supplementary Table 1, we show full original genera names output from QIIME to facilitate reproducibility.

- Supplementary Tables 6, 9 (check if others): too many parentheses (USA(...)) in the title?

Response: We removed unnecessary parentheses in the titles and added table footnotes as appropriate.

- Supplementary Figures 7-13: the heatmap cell width could be shrunk for improved readability

Response: We shrunk the heatmap cell width of relevant Supplementary Figures as suggested.

- Supplementary Figure 14: how about just using gray color scale here, and highlighting the Diarrhea cases with black (others darkgray). This would improve readability and highlight the interesting pattern more visibly. It is also mildly confusing that healthy state is marked with red now, as red color is often associated with health problems or warning signals; and diarrhea (blue) is lost somewhere in the middle of the figure; increased transparency (alpha) could be used for the error intervals, and the error intervals could be shaded with the same color than their corresponding lines. Now both cases have dark gray variation intervals, which slows down figure readability (or course if you switch to gray scale then this will remain so but even then it would be possible to consider different levels of shading to highlight the difference)

Response: We remade this Supplementary Figure as suggested. We want to expressly thank the reviewer for all of the constructive, thoughtful, and especially detailed comments regarding presentation and visualization of our meta-analysis results. We feel fortunate to have learned so many helpful tips about microbiome research that will serve us well into the future, and hope that sharing our findings and codes with the broader scientific community will enable the significant contributions of the reviewer to reach a broad audience as well.

REVIEWERS' COMMENTS:

Reviewer #2 (Remarks to the Author):

The authors have incorporated my review feedback appropriately in the revised manuscript, and I have no further suggestions for improvement.

Leo Lahti